# A Straightforward Method for the Isolation and Cultivation of *Galleria mellonella* Hemocytes

**DOI:** 10.3390/ijms232113483

**Published:** 2022-11-03

**Authors:** Joana Admella, Eduard Torrents

**Affiliations:** 1Bacterial Infections and Antimicrobial Therapies Group, Institute for Bioengineering of Catalonia (IBEC), The Barcelona Institute of Science and Technology (BIST), Baldiri Reixac 15-21, 08028 Barcelona, Spain; 2Microbiology Section, Department of Genetics, Microbiology and Statistics, Faculty of Biology, University of Barcelona, 643 Diagonal Ave., 08028 Barcelona, Spain

**Keywords:** nanoparticle, phagocytosis, *Galleria mellonella*, infection, cell culture

## Abstract

*Galleria mellonella* is an alternative animal model of infection. The use of this species presents a wide range of advantages, as its maintenance and rearing are both easy and inexpensive. Moreover, its use is considered to be more ethically acceptable than other models, it is conveniently sized for manipulation, and its immune system has multiple similarities with mammalian immune systems. Hemocytes are immune cells that help encapsulate and eliminate pathogens and foreign particles. All of these reasons make this insect a promising animal model. However, cultivating *G. mellonella* hemocytes in vitro is not straightforward and it has many difficult challenges. Here, we present a methodologically optimized protocol to establish and maintain a *G. mellonella* hemocyte primary culture. These improvements open the door to easily and quickly study the toxicity of nanoparticles and the interactions of particles and materials in an in vitro environment.

## 1. Introduction

*Galleria mellonella* is an alternative animal model of infection. Over the past few years, it has become quite popular due to its multiple advantages. The rearing and maintenance of the larvae of this species are not expensive and do not require specific facilities. The larval size ensures ease of manipulation and allows injection of precise doses of pathogens and drugs. *G. mellonella* is also a valuable model because it reduces the number of mammals used in research. Moreover, this insect presents an immune system that is similar to mammalian immune systems and can be divided into both cellular and humoral defenses [1,2].

Molecules such as opsonins and antimicrobial peptides or enzymatic processes such as melanization confer the humoral response. The synthesis of melanin during hemolymph coagulation shows the combined activity of both humoral and cellular immunity. Hemocytes, which mostly freely circulate in the hemolymph, are involved in the cellular response. Overall, these mechanisms contribute to the pathogen’s encapsulation and elimination [3,4].

At least eight types of hemocytes have been described in insects. However, not all insects present all classes. In *G. mellonella*, plasmatocytes and granulocytes are the most abundant cells and the only hemocytes capable of adhesion; they also carry out phagocytosis. Plasmatocytes are larger and circular, although their shape changes once they adhere to a surface by developing pseudopodia. These hemocytes have been implicated in nodulation. On the other hand, granulocytes are smaller and spherical. Their granular content is released when interacting with a foreign body. Spherulocytes, which are cells that present inclusions and seem to oversee some transport functions, can also be found. Oenocytoids are large cells mainly involved in the melanization process and the release of nucleic acids. Finally, prohemocytes are believed to be the precursors of all other types of hemocytes [5,6,7].

Cell cultures provide a wide range of useful applications. Primary cell cultures are those in which cells have been taken from original tissue, whereas subsequent cultures from cell multiplication are known as secondary cultures. Primary cultures more closely resemble the tissue of origin than cell lines, which is why they are appropriate model systems. Since the beginning of the 20th century, insect cells have been cultured but have encountered many obstacles. In 1962, Grace developed the first insect cell line by modifying a previously utilized type of culture media to more closely resemble insect hemolymph [8,9,10].

Establishing and maintaining a primary culture of *G. mellonella* hemocytes for an extended period of time is still a struggle due to its several difficulties. In addition, invertebrate cells generally proliferate slowly [11], and few studies on this subject can be found. However, the limitations and improvements described in *G. mellonella* and other invertebrate cell cultures helped us to move forward. Here, we report some methodological optimizations for the establishment and maintenance of *G. mellonella* hemocyte primary cultures.

The field of nanomedicine is becoming increasingly more popular, promising, and extensive every day. Nanoparticles and carriers made of all sorts of materials with multiple and diverse properties and targets can be found. Nanomedicine still faces several challenges even after more than two decades of study [12]. Once the particles are characterized, the first assays are performed in vitro. These models are much less complex than an actual in vivo organism but are essential to perform a first screening of compounds to evaluate their potential toxicity and effects. Most of these experiments are carried out in cell lines, and the use of primary cells is increasing. Cell culture allows the performance of biochemical assays and the staining and study of cell interactions, cellular uptake, and intracellular localization [13].

Primary *G. mellonella* hemocyte cultures can potentially allow the quick and easy study of nanoparticle toxicity and the interactions of particles and materials with cells, so we hope other researchers can test it. Moreover, after performing these sorts of experiments, we can move to the *G. mellonella* animal model and continue more studies in vivo.

## 2. Results and Discussion

### 2.1. Optimization of a G. mellonella Hemocyte Primary Culture

This work aimed to optimize and establish a straightforward methodology that could allow the use of *G. mellonella* hemocytes for further biological studies without requiring specific facilities. Here, as proof of concept, we examined their use by studying nanoparticle–cell interactions.

The literature contains different protocols for *G. mellonella* hemocyte isolation and use [14,15,16]. As a summary of such protocols, in our laboratory, we obtained hemocytes from larvae of approximately 250 mg. The larvae were sterilized in a solution of 70% ethanol before being anesthetized on ice. To obtain hemolymph, their tails were carefully cut in the more distal segment to avoid the disruption of the gut. Once the hemocyte cells were isolated and washed, they were quantified with trypan blue and incubated at room temperature with Grace’s supplemented insect medium with 2% penicillin–streptomycin, 2.5 µg/mL amphotericin B, and 10% fetal bovine serum added to the media (Figure 1).

Surprisingly, our results after following the literature [14,15,16] showed that almost all of the hemocytes isolated directly from larvae (Figure 2A) were nonviable after 24 h of isolation (Figure 2B) and completely dead after 6 days (see Figure 2C). Clearly, these apparently easy protocols described for the cultivation of *G. mellonella* hemocytes had some limitations, as a similar survival rate was expected during isolation and incubation. For this reason, we commenced hemocyte cultivation method optimization to increase their viability throughout the isolation methodology and protocol.

To increase hemocyte viability, we first attempted to optimize hemocyte cell culture using different media, supplements, and incubation conditions (Table 1). Cells were grown in DMEM/F12 (Dulbecco’s modified eagle medium nutrient mixture F12, Gibco) or Grace’s supplemented insect medium. In each experiment, the media was supplemented with 2% penicillin–streptomycin and 2.5 µg/mL amphotericin B [14]. In addition, we tested different percentages of FBS supplementation, different temperatures, and incubation under microaerophilic conditions (5% CO_2_).

Several of the different tests (examining culture media, temperature, and CO_2_ incubation) led to unsuccessful cultures with abundant cell death, with the highest cell viability of 15% (Table 1). Grace’s supplemented insect medium was established as the best medium for culturing hemocytes, as it gave higher cell viability. Regarding the temperature, it was observed that 37 °C was associated with a significant loss of cell viability (>95%). Dead cells looked darker than the healthy cells (data not shown), which are refractive and rounded [9]. Therefore, the temperature was established as 25 °C (room temperature) with an atmospheric CO_2_ concentration to maintain the physiological pH. FBS supplementation did not make such a large difference, and was optimized at 10%, which is the percentage commonly used for cell culture. After bibliographic investigation, our optimizations were in accordance with other insect cell cultures [9,15], although hemocyte viability was still low (≈15%) and unsuitable for further experiments.

Once the best hemocyte culture conditions were established, one of the main issues for cell maintenance was the early appearance of melanization, especially in areas with greater hemocyte density. Thus, the next step in our optimization was to prevent the melanization process by blocking phenoloxidase, a key enzyme in melanin synthesis.

With the addition of 0.6 mg/mL L-cysteine [10], melanization was inhibited entirely in our cultures. Other compounds, such as phenylthiourea (PTU) [17] or glutathione [10], are also useful inhibitors, as described by others. During the hemolymph collection process, melanization was prevented by keeping the samples on ice, so we added L-cysteine only as a media supplement in all hemocyte cell cultures. With this improvement, we tested cell viability again in both Grace’s medium and DMEM to validate that our chosen conditions were suitable, with hemocyte viability increased by 15% in one day (Figure 3). Again, we confirmed that the use of DMEM and 37 °C damaged the cells, as also seen in Table 1. Activation of the melanization enzymatic cascade in *G. mellonella* hemocyte cultures is a well-known problem [18], and growing the cells without the use of melanization inhibitors is a major challenge (Figure 4A). This observation has also been reported in cell cultures of other invertebrates that possess hemocytes [18,19,20].

However, another issue observed was the presence of cell aggregates (Figure 4B) during cultivation. In vitro, hemocytes tend to clump due to cell instability [21]. Thus, we tested the use of an anticoagulant solution [22] during the hemocyte washing steps to help reduce cell clumping. Furthermore, we observed that mixing cells with the anticoagulant solution added directly to the cell culture medium containing FBS induced high aggregation rates. To avoid such aggregation, a period of 30 min was added before the cells settled in FBS-free medium, and then, 10% FBS was added to the cell culture [18]. This strategy reduced the clumping of cells in the plate, helping the attachment of hemocytes to the wall surface and improving hemocyte appearance and viability (Figure 4C,D). Thus, it was found that avoiding cell aggregation was crucial for establishing hemocyte cultures. The final optimal protocol for culturing hemocytes in vitro is summarized in Figure 1.

### 2.2. Quantification of Hemocyte Viability

While optimizing the conditions of our cell cultures, we faced the challenge of finding an optimal method for monitoring hemocyte viability over time. As summarized in Figure 5, different procedures were investigated. In adherent microtiter plate cultures, cells are usually dissociated enzymatically or mechanically, so viability can be easily measured with trypan blue. Both trypsin (Fisher Scientific, Waltham, MA, USA) and TrypLE Express (Fisher Scientific), standard enzyme solutions in cell biology laboratories, were tested unsuccessfully. Hemocyte detachment with a cell scraper was also attempted, but none of these procedures were well-suited for hemocytes, as they appeared to be both ineffective and highly damaging (data not shown).

Other methodologies were investigated to measure the evolution of hemocyte culture over time. Fast live cell assays were also performed, which utilized viability reagents such as methylthiazol tetrazolium (MTT) (Sigma-Aldrich, St. Louis, MO, USA) or PrestoBlue (Fisher Scientific). Both assays are colorimetric and measure cell metabolic activity. The MTT assay is based on the reduction of the tetrazolium salt (MTT) by mitochondrial succinic dehydrogenases in viable cells. This yields purple formazan crystals that need to be solubilized in isopropanol [23]. The absorbance can then be measured with a spectrophotometer. The PrestoBlue assay is similar to the MTT assay, but in this case, the reagent exhibits a color change due to the conversion of resazurin to resorufin, a highly red fluorescent compound generated by metabolically active cells. This produces a shift in the fluorescence, allowing for quantification by either a fluorometric or spectrophotometric approach [24].

During the initial steps of methodological optimization in this work, the number of active cells was low because the MTT assay did not appear work in those cultures, giving a low absorbance signal. In the case of PrestoBlue, fluorescence is the preferred detection method, as it is more sensitive than absorbance. As we improved our hemocyte cultures, the fluorescence signal also increased, and a change in color was directly visible and more evident in the first days of culture.

All of these assays are usually performed in a 96-well plate, but we noticed that the cells were in worse shape under these conditions than when the cells had more space, as in a 24-well microtiter plate (see Section 3.2). Another limitation of the PrestoBlue assay was that we could not determine the percentages of active and inactive cells, as we only had the fluorescence value for each measurement.

For all of these reasons, we tested the live/dead staining method (Figure 5). The dyes (SYTO 9 and propidium iodide) were added to the respective wells. Cells with intact membranes stained fluorescent green, whereas cells with a damaged membrane (dead or dying cells) stained fluorescent red [25]. To carry out this procedure, more than 60 images were taken per well per day to cover almost the entire surface of each well. These images were later processed by ImageJ FIJI to count the proportion of each cell type (Figure 6). This was the most accurate method to determine culture viability on each day of measurement while also growing the cells in a larger microtiter plate.

In Figure 6, Day 0 refers to hemocyte viability in the initial culture measured 4 h after hemocyte extraction (Figure 1). Large differences in hemocyte viability were observed between our very first results (Table 1) and those in this graph. If we gather all viability data from process optimization, L-cysteine supplementation provided a 17% increase in cell viability from 15% (Table 1) to 32.38% on Day 1 (Figure 3). Then, the use of the anticoagulant solution was crucial for increasing cell viability to yield 50% live cells, which is the value represented in Figure 6. One aspect to consider in this graph is that on Day 0, half of the cell population was already dying cells despite the hemocytes having a perfect appearance and morphology. Knowing this, the loss of viability reported after one week is not that large (30%), so these data start on Day 0 from just 50% viability. If we look at the line graph, the percentages of live cells counted on all measured days are larger than those of dead cells. This is because the number of cells counted each day decreased over time due to cell death.

*G. mellonella* larvae are small, so hemolymph needs to be extracted and pooled from several animals. When hemolymph is exposed to air, immediate blood coagulation takes place, as is the case for many insects [26]. That is why the use of an anticoagulant solution is key for avoiding cell lysis, degranulation, and clotting, all of which occur when the harvested cells are suddenly introduced in a saline buffer [27]. The use of anticoagulant solutions when working with insects and other invertebrates is not a new idea, as it was used in the 1980s [28]. However, many recent publications do not include this step. Without the prevention of melanization and cell clotting, our cultures would have never become stable and viable enough for further experiments.

### 2.3. Hemocyte Functionality and Activity

Our next step was to evaluate the ability of the hemocytes to perform phagocytosis after long-term culture, which is a good indicator that they are alive and active. For this, a suspension of fluorescent-tagged *Mycobacteroides abscessus* and fluorescent nanoparticles [29] were added to different cultures and visualized after 24 h under a confocal microscope to evaluate the cell response to these stimuli. It was observed that in both 2- and 6-day cultures, there were many active hemocytes that were able to phagocytize the bacteria and nanoparticles. In particular, after Imaris software (version 7.4.2, Abingdon, UK) reconstruction, the bacteria and nanoparticles were clearly observed to be internalized by the hemocytes (Figure 7).

Next, we were interested in evaluating the activity of the cultured hemocytes by measuring changes in the gene expression of different *G. mellonella* immune-relevant genes. For this, *G. mellonella* RNA was extracted from hemocytes in culture three hours after the cells had been stimulated with *Mycobacteroides abscessus* (Figure 8). A methodological limitation in this procedure was the low RNA concentrations obtained from the primary cell culture. Thus, to increase RNA yield, many larvae were needed in the first step to increase the number of initial cells; still, the RNA concentrations were low. Among the analyzed genes, there were opsonins (hemolin), antimicrobial peptides (gloverin), and enzymes (NOS, GST). RNA was retrotranscribed, and real-time PCR was performed (see Section 3.5). As shown in Figure 8, we observed the induction of gene expression of all of the above genes in response to bacterial inoculation.

The highest induction (10.84 times) was seen for hemolin, an opsonin exclusive to Lepidoptera. Hemolin is part of the immunoglobulin family and is able to recognize and bind to different pathogen-associated molecular patterns (PAMPs) on the bacterial surface. Hemolin also participates in the immune response by agglutinating bacteria and inhibiting hemocyte aggregation in vitro. Upregulation of its transcription after bacterial infection has been previously described in Lepidoptera [30,31,32]. Gloverin is a glycine-rich antimicrobial peptide that is also specific to lepidopteran insects. It seems to have an effect on filamentous fungi and both Gram-positive and Gram-negative bacteria [33]. Upregulation of its transcription has already been reported in larvae that were challenged for 24 h with different species of *Bacillus* [34]. In *Manducta sexta*, another Lepidoptera species, gloverin has also been shown to be induced both transcriptionally and at the protein level by different kinds of microorganisms, including both Gram-positive and Gram-negative bacteria [35]. In our case, a 5.19-fold induction was observed.

Regarding the analyzed enzymes, nitric oxide synthase (NOS) produces nitric oxide, which inhibits bacterial growth with other ROS. It has been reported that bacterial infections significantly increase the transcription levels of ROS-related genes [36]. NOS mRNA levels were induced in a tissue-dependent manner in *M. sexta* previously infected with *Photorhabdus luminescens*. This and previous studies suggest a relevant role of NOS in the immune system of several insects, including *G. mellonella* [37]. In our experiment, we found a fold change of 7.85.

Finally, glutathione S-transferases (GSTs) are an extensive and diverse family of detoxification enzymes found in most organisms. GSTs help to protect cells from oxidative stress and play a role in detoxification [38]. Foreign infections in host cells cause oxidative stress and the production of ROS, which disturbs the balance in antioxidant defenses. It has been observed in other insects that ROS can induce the expression of some GSTs [39]. *G. mellonella* larvae infected with *Bacillus thuringiensis* showed increased GST activity in their midgut. As other studies have shown, this suggests the involvement of GST in the elimination of ROS in an early stage of infection [40]. In our hemocytes, GST was induced 8.5-fold. Overall, our results showed an in vitro response to the bacterial inoculum by the cultured hemocytes.

Some recent publications [17,41] describe the use of *G. mellonella* hemocytes primary cultures for different purposes, as described in this work. These authors have successfully performed short-term experiments with their cultures by following a different methodology. However, we were interested in cultivating and maintaining the hemocytes for several days, which is the main difference from the published previous work.

Therefore, in this study, we established and maintained an optimized *G. mellonella* hemocyte primary culture and proved their activity in vitro. This straightforward, inexpensive, and quick methodology has potential for studying the toxicology of nanoparticles and their interactions with cells, as well as other functional studies involving this cell type. However, much work is still needed in the field of invertebrate hemocyte primary cell culture, particularly in *Galleria mellonella*. For this, we hope future work will be able to further increase the stability and viability of hemocytes in culture.

## 3. Materials and Methods

### 3.1. Galleria mellonella Maintenance and Hemolymph Extraction

*G. mellonella* larvae were fed an artificial diet (15% corn flour, 15% wheat flour, 15% infant cereal, 11% powdered milk, 6% brewer’s yeast, 25% honey, and 13% glycerol) and reared at 34 °C in darkness. Between 50 and 60 larvae of approximately 250 mg (35–40 days old at latest larval stage) were swabbed with 70% ethanol and anesthetized on ice for at least 10 min. Larvae tails were cut off with a size 23 sterile surgical blade, and their hemolymph was collected into Eppendorf tubes on ice to avoid melanization. Hemolymph was pooled from at least 10 larval groups, added to 100 µL of anticoagulant solution (26 mM sodium citrate, 30 mM citric acid, 100 mM glucose, and 140 mM NaCl, pH = 4.11) [22], and centrifuged at 200× *g* and 4 °C for 5 min. Pellets were resuspended in 50 µL of anticoagulant solution and washed three times. Finally, 10 µL of the solution was mixed with 10 µL of trypan blue (Sigma-Aldrich) to determine the cell concentration with a hemocytometer (Figure 1). The average cell density was approximately 5 × 10^6^ cells/mL.

### 3.2. Maintenance of a Hemocyte Primary Cell Culture

With our optimized procedure, the washed cell suspension was added to a polystyrene 24-well flat bottom microtiter plate (SPL Life Sciences, Busan, Korea) with 900 µL of Grace’s insect medium (Gibco, Billings, MT, USA) supplemented with 2% penicillin–streptomycin (Gibco), 2.5 µg/mL amphotericin B (Gibco), and 0.6 mg/mL L-cysteine (Sigma-Aldrich). After approximately 30 min, when the cells had settled, 100 µL of fetal bovine serum (FBS) (Gibco) was added. Cultures were maintained in the dark at 25 °C. The medium was replaced every 2–3 days, and the cells were observed regularly under an inverted fluorescence microscope (ECLIPSE Ti−S/L100, Nikon) coupled with a DS-Qi2 camera (Nikon) using the 10×/0.25 Ph1 objective.

### 3.3. Quantification of Hemocyte Viability in Culture

#### 3.3.1. PrestoBlue Cell Viability Assay

A total of 10^5^ hemocytes/well were placed in a polystyrene 96-well flat bottom microtiter plate (Corning Costar) with 200 µL of Grace’s supplemented insect medium. Once the cells had attached to the well surface, the media was removed, and a solution containing 10 µL of PrestoBlue (Invitrogen) and 90 µL of media was added to each well. After a 2-h incubation, the fluorescence was measured according to the PrestoBlue Protocol in a Spark multimode microplate reader (TECAN). The gain was adjusted to the optimal value of 65. The Z-position was established as 30,000 μm, and the integration time was set to 40 μs. This procedure was performed on different days to measure cell viability over time. The results were plotted with GraphPad Prism 9.0 software (San Diego, CA, USA).

#### 3.3.2. Live/Dead Cell Viability Assay

A total of 10^5^ cells/well were used to study hemocyte culture viability for one week in a polystyrene 24-well flat bottom microtiter plate (SPL Life Sciences). For this purpose, a Live/Dead Viability Kit (Invitrogen) was used according to the manufacturer’s protocol to dye both active and damaged cells by taking multiple images of the well with an inverted fluorescence microscope (ECLIPSE Ti−S/L100, Nikon) coupled with a DS-Qi2 camera (Nikon) using the 20×/0.45 Ph1 objective with GFP and Texas Red filters for green and red fluorescence, respectively. Cells were counted with ImageJ FIJI (version 1.52p), and the results were plotted with GraphPad Prism 9.0 software. The statistical analyses were also performed with GraphPad Prism 9.0 software.

### 3.4. Hemocyte Imaging by Confocal Microscopy

Different cultures were set in 35 mm culture confocal cell dishes (VWR). Approximately 10^4^ cells were placed in 2 mL of culture medium. Between 2 and 6 days later, 10 µL of red fluorescent rhodamine nanoparticles (100 µg/mL) [29] and 10 µL of a smooth suspension of *Mycobacteroides abscessus* (ATCC_19977) transformed with the plasmid pFPV27 encoding a constitutive GFP [42] were added to each culture and incubated at room temperature in darkness for 24 h. Cells were stained with 300 nM DAPI (Invitrogen, Waltham, MA, USA) and at 5 µg/mL FM^TM^ 4–64 (Invitrogen). After approximately 30 min, hemocytes were observed under an LSM 800 confocal laser scanning microscope (Zeiss, Aalen, Germany) with a 63×/1.4 oil objective. Images were analyzed with ImageJ FIJI (version 1.52p) and Imaris Cell Imaging Software (version 7.4.2; Abingdon, UK).

### 3.5. RNA Extraction, Reverse Transcription, and Real-Time PCR

To measure the expression of some *G. mellonella* immune-relevant genes, approximately 7.5 × 10^5^ cells/condition were placed in a polystyrene 6-well flat bottom microtiter plate (Caplugs Evergreen, Caplugs, CA, USA) with 2 mL of the optimized supplemented medium (see Section 3.2). After 72 h, 100 µL of 1× PBS pH 7.5 (Fisher Scientific) and 10^6^ CFUs (100 µL) of *M*. *abscessus* were added to the respective cell cultures and incubated for 3 h. A cell scraper was used to detach the cells. Cells were recovered with 200 µL of 1× PBS and homogenized with 300 µL of lysis buffer in a 20 G syringe. RNA was purified using the GeneJET^TM^ RNA Purification Kit (Fisher Scientific) according to the manufacturer’s protocol. TURBO™ DNase (Fisher Scientific) was used to remove DNA contamination, and a DNA absence test was performed by PCR for verification.

For cDNA synthesis, RNA was quantified using a NanoDropTM 1000 spectrophotometer (Fisher Scientific). Reverse transcription PCR was carried out with Maxima Reverse Transcriptase (Fisher Scientific) to obtain cDNA, and from that, quantitative real-time PCR (qRT-PCR) was performed using PowerUp^TM^ SYBR^TM^ Green Master Mix (Applied Biosystems) according to the manufacturer’s instructions in a StepOnePlus^TM^ Real-Time PCR System (Applied Biosystems, Foster City, CA, USA). All qRT-PCRs used specific gene primers (Hemolin-For, 5′-CCCGAAGACGCTGGTGAATA-3′; Hemolin-Rev, 5′-CGCACGTTCATTTGCTGTTC-3′; Gloverin-For, 5′-AGATGCACGGTCCTACAG-3′; Gloverin-Rev, 5′-GATCGTAGGTGCCTTGTG -3′; NOS-For, 5′- ATGAAGGTGCTGAAGTCACAA -3′; NOS-Rev, 5′-GCCATTTTACAATCGCCACAA-3′; GST-For, 5′-GACAGAAGTCCTCCGGTCAG -3′; NOS-Rev, 5′-TCCGTCTTCAAGCAAAGGCA-3′; 18S-For, 5′-ATGGTTGCAAAGCTGAAACT-3′; 18S-Rev, 5′-TCCCGTGTTGAGTCAAATTA-3′). The 18S ribosomal RNA gene was used as an internal standard because its expression is vital and constant in *G. mellonella*. For each sample, three replicates were performed. The results were analyzed using the comparative Ct (cycle threshold) method (∆∆Ct) and plotted with GraphPad Prism 9.0 software as previously described [43]. The statistical analyses were also performed with GraphPad Prism 9.0 software.

## Figures and Tables

**Figure 1 ijms-23-13483-f001:**
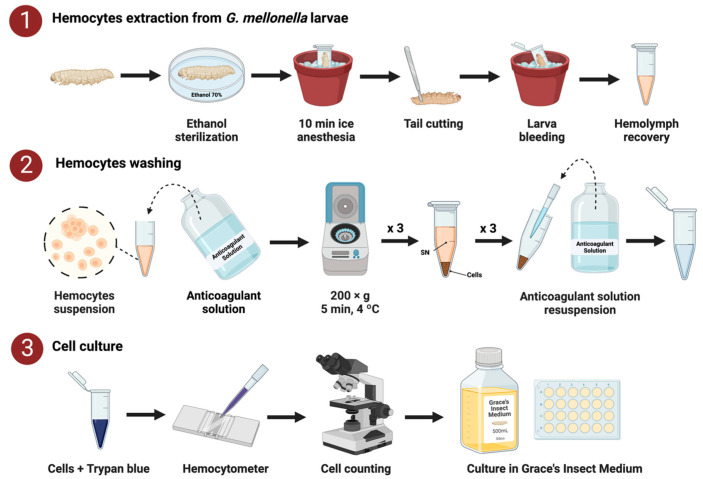
General methodology used to establish a primary culture of *G. mellonella* hemocytes. The process included the recovery of larval hemolymph (**1**), centrifugation, washing (**2**), and quantifying and seeding the hemocytes in supplemented culture medium (**3**) (see Section 3.1 and Section 3.2). Created with www.Biorender.com (accessed on 28 September 2022).

**Figure 2 ijms-23-13483-f002:**
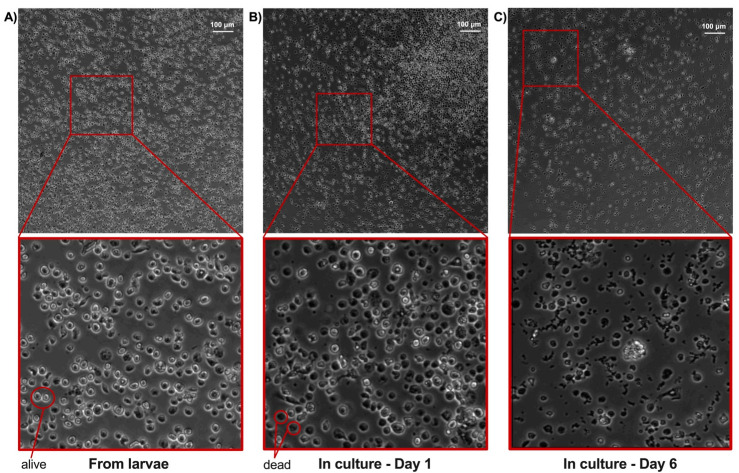
Evolution of the hemocyte primary cell culture. The red square indicates a magnification of the indicated image. Live cells appear refractive, whereas dead cells can be identified by their darker color (see arrows). (**A**) Hemocytes directly extracted and washed from the hemolymph of larvae. (**B**) Hemocytes in Grace medium on Day 1 of culture. (**C**) Hemocytes in Grace medium on Day 6 of culture. Substantial cell debris can be seen in the amplified images.

**Figure 3 ijms-23-13483-f003:**
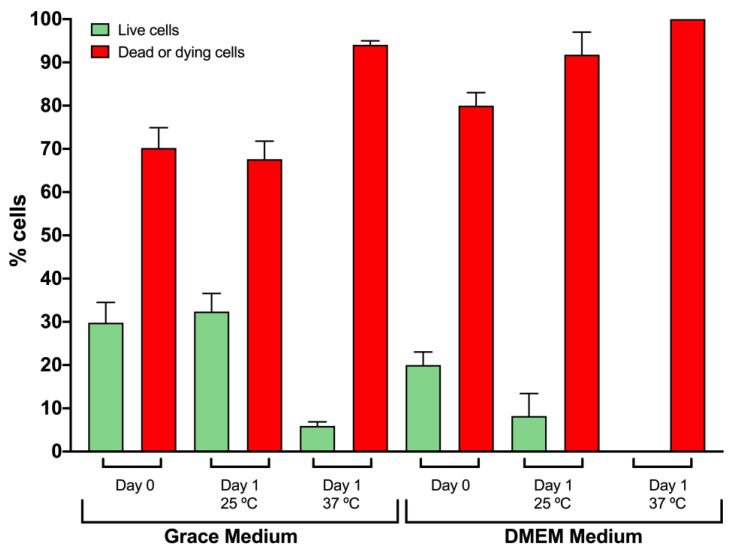
Viability of hemocytes in culture after adding L-cysteine as a new media supplement in different media and at different temperatures. Three independent experiments were performed. The error bars indicate a positive standard deviation.

**Figure 4 ijms-23-13483-f004:**
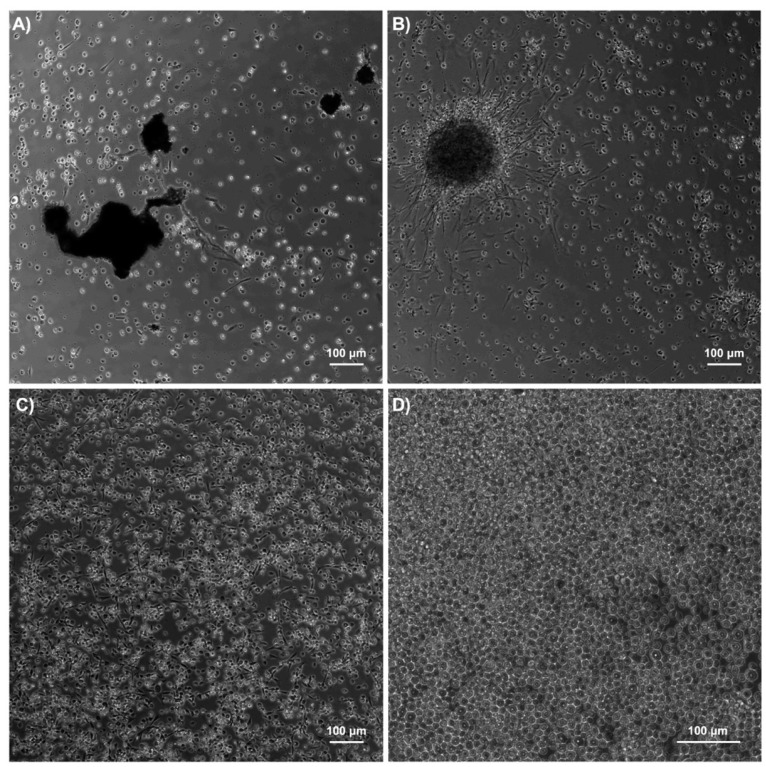
Microscopic images of hemocyte cultures. (**A**) Culture with melanized areas shown in black. (**B**) Culture with a large cell aggregate. (**C**) Culture with cells attached to the well. (**D**) Optimized culture where there are no clumps or melanization due to the use of both L-cysteine and the anticoagulant solution. All images belong to different cultures on Day 1 of observation. Images (**A**–**C**) were taken at a magnification of 10×, while image (**D**) was taken with a magnification of 20×.

**Figure 5 ijms-23-13483-f005:**
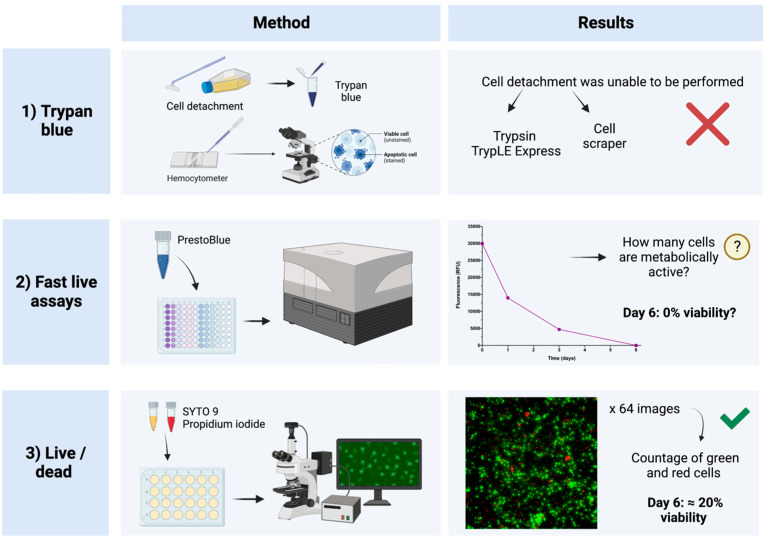
Summary of the different cell viability methods tested. Created with www.Biorender.com (accessed on 28 September 2022).

**Figure 6 ijms-23-13483-f006:**
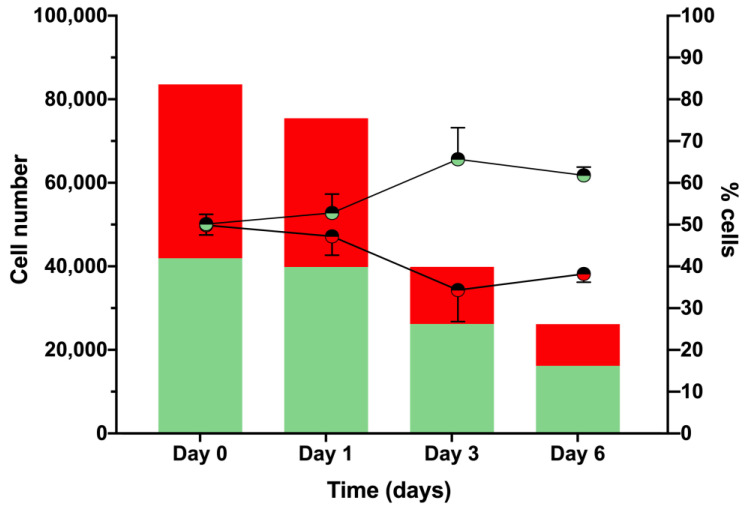
Hemocyte viability in the optimized culture over time. Cell survival was measured over 7 days. The left Y-axis indicates the number of cells counted from multiple images taken each day, and the results are shown in the bar graph. The right Y-axis indicates the percentage of each cell type, represented as a line graph. Two independent experiments were performed. The error bars in the line graph indicate a positive standard deviation. Green and red colors represent live and dead cells, respectively.

**Figure 7 ijms-23-13483-f007:**
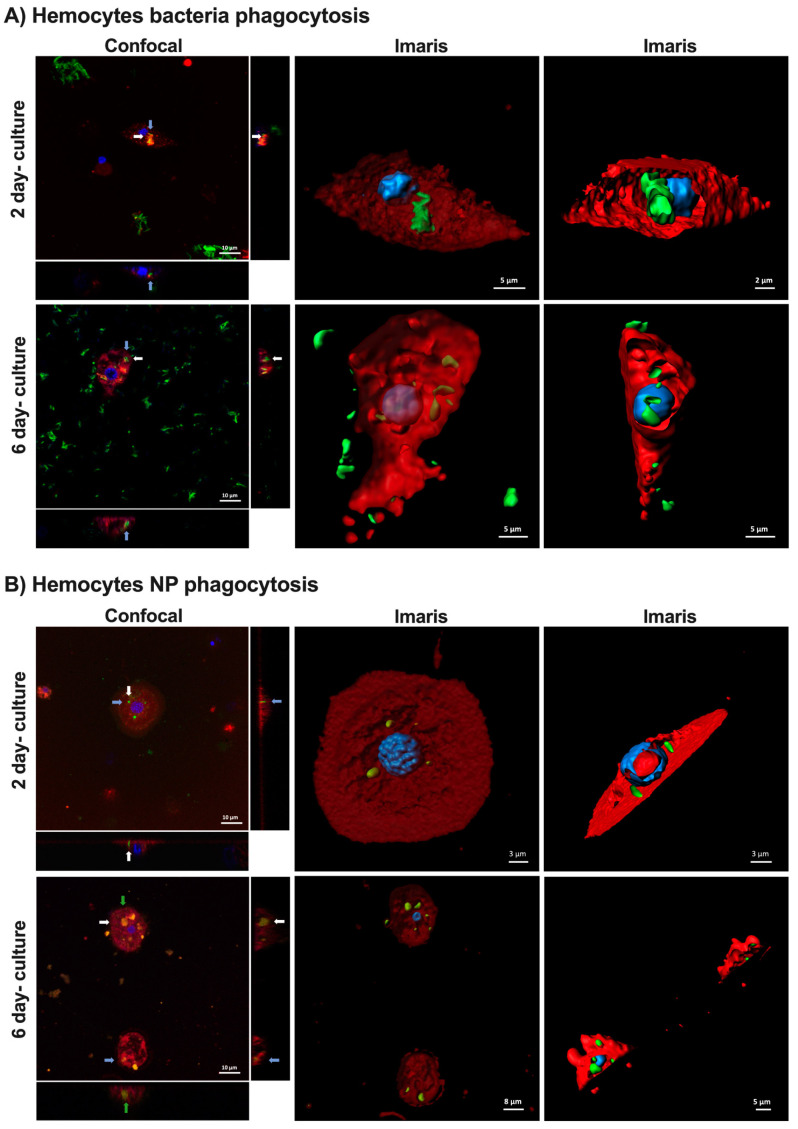
Hemocyte phagocytosis of bacteria and nanoparticles in 2- and 6-day cultures. (**A**) Hemocytes (cells in red with blue nuclei) with phagocytosed bacteria (green) in both the confocal and Imaris reconstruction images. (**B**) Hemocytes (cells in red with blue nuclei) with phagocytosed nanoparticle aggregates (green) in both the confocal and Imaris reconstruction images. The same nanoparticle in the orthogonal and sumstack view is shown with the same arrow color.

**Figure 8 ijms-23-13483-f008:**
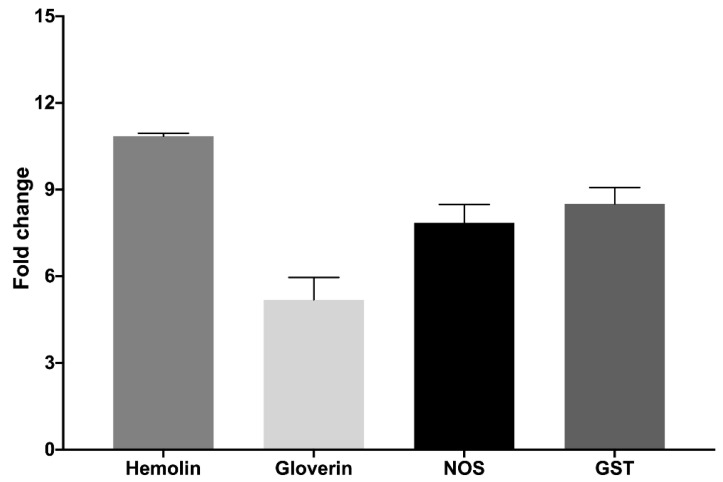
Fold changes in *G. mellonella* immune-relevant gene expression determined by qRT–PCR of hemocytes in culture stimulated with *M. abscessus* compared with unstimulated hemocytes. The 18S ribosomal RNA gene was used as an internal standard gene. The values shown are the average from one experiment performed in triplicate. The error bars indicate a positive standard deviation.

**Table 1 ijms-23-13483-t001:** Test of different culture conditions (media, FBS supplementation, temperature, and CO_2_ presence) during the cell culture optimization process and the effect of each variable on cell viability. An approximation of the cell viability was determined by examining the appearance of the cells on Day 1 of cell culture in a series of 3–5 different experiments.

	Medium	Fetal Bovine Serum (FBS)	Temperature	CO_2_	Day 1 Cell Viability
1	DMEM	10% (*v*/*v*)	37 °C	Yes	0%
2	DMEM	10% (*v*/*v*)	25 °C	No	<10%
3	Grace	10% (*v*/*v*)	37 °C	Yes	<5%
4	Grace	15% (*v*/*v*)	37 °C	Yes	<5%
5	Grace	20% (*v*/*v*)	37 °C	Yes	<5%
6	Grace	10% (*v*/*v*)	37 °C	No	<5%
7	Grace	10% (*v*/*v*)	25 °C	No	≈15%
8	Grace	15% (*v*/*v*)	25 °C	No	≈15%
9	Grace	20% (*v*/*v*)	25 °C	No	≈15%

## Data Availability

Not applicable.

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
