# Peer review of "A Straightforward Method for the Isolation and Cultivation of Galleria mellonella Hemocytes"

_ijms, 2022, doi:10.3390/ijms232113483_

Round 1

Reviewer 1 Report

Dear Authors,

Thank you for the opportunity to read more about your work.  I think that the presented experimental protocol is essential, and I appreciate their hard work. The results are really interesting.

Through reading this interesting manuscript, here are a few highlights about issues that would cause concern:

Line 87 and 337 In addition to the body weight of the larvae, did the authors also pay attention to the developmental stage? Hemocytes in different larval stages may behave slightly differently in cell culture.

From how many insects was the hemolymph collected for one sample? What was the average cell density in one sample?

Overall, how many replications were each experiment done to optimize the culture method and measure cell viability? There are no standard deviations in the tables and graphs, which looks as if each experiment was performed only in one repetition

Line 88 and 338 After rinsing the larvae in ethanol, were they then rinsed in ultrapure water? Small amounts of ethanol can enter the hemocyte cultures and affect their development

Line 120 Do I understand correctly that cells were cultured in one or the other medium? If so, it might be better to change 'and' to 'or' so as not to imply that the cells were grown in the two media mixed together

Line 148 In the literature (the work of the team of M.I. Boguś and A.K. Wrońska) there is also a lot of information about the use of PTU (phenylthiourea) as a substance preventing melainization. Did the authors also plan to use this method?

Fig 4. Please add in the description of the figure the information at what magnification the photos were taken. There is also no information on which day of cell culture the photos were taken. If I understand correctly, the observations of the hemocyte cultures were conducted for 6 days. Maybe it is worth posting photo documentation of hemocytes day by day? Did the authors try to find individual cell subpopulations in the photos? Perhaps it is worth adding photos of individual cells from the given classes in a magnified view to show how they looked like during the experiment.

Table 1, Fig 3. What method was the viability of the cells assessed in culture? Only chapter 2.2 describes the optimization of the method for measuring haemocyte viability. If a non-optimal method of assessing viability was used in Section 2.1, it could have distorted the results of the experiment

Fig 7. Did the authors manage to identify what classes of hemocytes were capable of phagocytosis and what class of cells are shown in the photos?

Line 280. The name of the bacteria appears for the first time in the text, so please provide the full name.

Lines 335-336. Has the composition of food for insects been developed by the authors or taken from the literature?

Line 383 Were rhodamine nanoparticles purchased commercially, if so, from which manufacturer?

Author Response

Dear Sir/Madam,

Thank you for the thorough review of our paper and the opportunity to submit a revised version this second time. Again, we much appreciate the reviewer’s constructive comments on our manuscript.

Our responses are detailed below in red and yellow in the highlighted manuscript.

With the manuscript changes detailed below and our answers to the reviewer’s comments, we hope you will now find the revised version of our manuscript acceptable for publication.

Sincerely,

Dr. Eduard Torrents

Reviewer 1

Dear Authors,

Thank you for the opportunity to read more about your work.  I think that the presented experimental protocol is essential, and I appreciate their hard work. The results are really interesting.

Through reading this interesting manuscript, here are a few highlights about issues that would cause concern:

Line 87 and 337 In addition to the body weight of the larvae, did the authors also pay attention to the developmental stage? Hemocytes in different larval stages may behave slightly differently in cell culture.

We appreciate the reviewer for pointing this out. Larvae were about 35-40 days old, in the latest larval stage. This was decided to obtain the maximum hemolymph volume while using the smallest number of insects possible. Therefore, smaller larvae in early developmental stages were not considered. This information was added as a new sentence on page 11 and line 353.

From how many insects was the hemolymph collected for one sample? What was the average cell density in one sample?

Thanks for this observation. To study the maintenance and viability of hemocytes in 24 well-plates, the number of insects used was between 50-60 larvae. The average cell density was approximately 5x106 cells/ml when using the optimized protocol. Information has been added in the manuscript in section 4.1, lines 353 and 362.

Overall, how many replications were each experiment done to optimize the culture method and measure cell viability? There are no standard deviations in the tables and graphs, which looks as if each experiment was performed only in one repetition

We thank the reviewer for pointing this out. Concerning Table 1, the viability column is a summary of 3-5 different experiments, which were performed at the very beginning of our project. A deviation does not appear in the table as Day 1 viability was calculated only as an estimation by observation. To clarify this point, we added more information on page 4, line 132. Also, please check the answer to one of your questions about the methodology used to generate this Table 1.

About Figure 3, we apologize for missing the statistics before. These experiments were performed in triplicates. The figure and figure legend have been modified accordingly. Please, see on page 5 the new figure and figure legend 3.

In Figure 6, the experiment with the optimized protocol was performed twice. The statistics had not been added before due to the complexity of the plot being a double graph. A new figure has been made, adding only the standard deviation bars in the line graph. The bar graph, which describes the number of cells, was quite similar, and we had to leave it like it was to make more accessible the visualization and understanding of the plot. Please, see on page 8 the new figure and figure legend 6.

In Figure 8, the values represented belong to one experiment performed in triplicates. The error bars indicate a positive standard deviation. Please, see the added information on figure legend 8, page 10, line 303.  

All statistics analyses were performed with Graph GraphPad Prism 9.0 software. Please, see the information added as a sentence in section 4.3.2 (line 406) and section 4.5 (line 448).

Line 88 and 338 After rinsing the larvae in ethanol, were they then rinsed in ultrapure water? Small amounts of ethanol can enter the hemocyte cultures and affect their development

We thank the reviewer for this observation. Larvae were not rinsed with water after ethanol sterilization. Reading through the literature, we saw many papers that described the rinsing of larvae with only ethanol.  Furthermore, it should be noticed that hemolymph was washed three times before hemocytes were added into the medium, so probably small amounts of ethanol would have already been excluded through the process. However, it is an excellent suggestion to conduct future experiments.

Line 120 Do I understand correctly that cells were cultured in one or the other medium? If so, it might be better to change 'and' to 'or' so as not to imply that the cells were grown in the two media mixed together

Thanks for pointing this out. Cells were cultured in one or the other medium. Please, see the modification on the manuscript on page 4, line 124.

Line 148 In the literature (the work of the team of M.I. Boguś and A.K. Wrońska) there is also a lot of information about the use of PTU (phenylthiourea) as a substance preventing melanization. Did the authors also plan to use this method?

We appreciate this observation. When trying to prevent melanization, we thought of different methods. Still, since the use of L-cysteine was tested first and it avoided melanization perfectly, we decided to continue with this substance, and we never tried phenylthiourea. In any case, we agree that it is another possibility that has worked for different authors, and indeed we need to test this compound in our experiments. We have added this suggestion as an alternative in our manuscript, please, see an extra sentence on page 5, line 153.

Fig 4. Please add in the description of the figure the information at what magnification the photos were taken. There is also no information on which day of cell culture the photos were taken. If I understand correctly, the observations of the hemocyte cultures were conducted for 6 days. Maybe it is worth posting photo documentation of hemocytes day by day? Did the authors try to find individual cell subpopulations in the photos? Perhaps it is worth adding photos of individual cells from the given classes in a magnified view to show how they looked like during the experiment.

We kindly appreciate the reviewer for making these suggestions. All images belong to different experiments, but all were from Day 1 of culture. All photos were taken at 10×, except image D), which was taken at a magnification of 20×. Please, see the modification in figure legend 4 (page 6, line 188).

We want to show the differences between those two conditions and time. Figure 2 shows the difference between cultures on the first and last days of observation (Day 6). At that point, cultures were not optimized, so Day 6 had a lot of mortality. After the optimizations, cell death and debris were reduced, so we did not consider adding a photo documentation of hemocytes daily. Moreover, our purpose in this figure was only the representation of how our main limitations (melanization and cell aggregation) were seen in the cultures and how these cultures improved in appearance once we used the optimal methodology. This is also why we have not to dig into the finding and identification of the different hemocyte populations.

Table 1, Fig 3. What method was the viability of the cells assessed in culture? Only chapter 2.2 describes the optimization of the method for measuring haemocyte viability. If a non-optimal method of assessing viability was used in Section 2.1, it could have distorted the results of the experiment

Thanks for pointing this out. In Table 1, as mentioned in its description, the cell viability was only estimated by observation in different experiments. These experiments were performed at the beginning of this project when we still had not optimized a technique for measuring hemocyte viability. Therefore, this is only an approximation of how the cell cultures looked during direct counting and visualization. When we found our optimal technique (Live/Dead methodology), these measurements were repeated with the addition of L-cysteine (Figure 3). Therefore, in Figure 3 we can reaffirm that the chosen mediums and conditions were the most suitable for hemocytes culture.

Fig 7. Did the authors manage to identify what classes of hemocytes were capable of phagocytosis and what class of cells are shown in the photos?

We appreciate the reviewer for this suggestion. As mentioned in the introduction (page 1, line 42), granulocytes and plasmatocytes are the only types of G. mellonella hemocytes capable of performing phagocytosis. Our purpose was never to identify and observe certain classes of hemocytes, as we were interested in the general cultivation of hemocytes. In any case, the significant percentage of the G. mellonella hemocyte population corresponds to those related to phagocytosis, as oenocytoids and spherulocytes represent only a minor percentage.

Line 280. The name of the bacteria appears for the first time in the text, so please provide the full name.

We agree. The full name has been added to the text. Please, see the modification on line 274.

Lines 335-336. Has the composition of food for insects been developed by the authors or taken from the literature?

We appreciate this observation. The food composition was done in our laboratory (as described clearly in the manuscript in section 4.1) by the same entomology research group that provided our laboratory with Galleria mellonella larvae some years ago. Since then, we have continuously fed the larvae with the same recipe, obtaining good results with their rearing and maintenance in our laboratory.

Line 383 Were rhodamine nanoparticles purchased commercially, if so, from which manufacturer?

We appreciate this question. The rhodamine nanoparticles have not been purchased commercially. They were available in our laboratory as they were used for other experiments. We have published the synthesis of this nanoparticle and have already cited how it was synthesized in reference 29 of our manuscript. See the sentence on page 12, line 401.

Reviewer 2 Report

Comments on the manuscript entitled: „A straightforward method for the isolation and cultivation of Galleria mellonella hemocytes” by Joana Admella and Eduard Torrents (Manuscript ID: ijms-1995811)

General comments

11. The Authors cited only some selected papers published on in vitro insect hemocytes’ culture. And yet there are papers in which primary cultures of G. mellonella hemocytes have been successfully used and medium composition as well as culture conditions have been provided. Please, see e.g. Bogus et al. (Bull Entomol Res 2017, 107, 66-76), Kaczmarek et al. (Int. J. Mol. Sci. 2022, 23, 5204) and many papers published by this team. The Authors should at least compare their own findings with those published already on G. mellonella hemocytes’ primary cultures.

22. In vitro evaluation of the toxicity of various compounds and structures (e.g. nanoparticles) to eukaryotic cells can successfully be performed using mammalian cell lines. The use of in vitro cultured insect hemocytes for this purpose can present many problems (including interpretation of results) even when using primary hemocyte culture, mainly because they are very sensitive, labile cells that dye fast.

33. There is no information about the performed statistical analysis of the results. 

Some other points

Line 19  - “… in an in vivo environment”? Is it correct?

Line 88 and Line 338 - If the tails of the larvae were cut off for hemolymph collection, how was the contamination of the hemolymph with intestinal material avoided? The hindgut must have been broken when the tail was cut off. Could the presence of such contaminants, as foreign bodies for hemocytes, cause melanization and/or aggregation of hemocytes? Usually, the hemolymph is collected by incising (not cutting off) one of the last proleg.

Lines 148-149 – Why was L-cysteine added only during the suspension phase in the medium and not during the hemolymph collection? Usually, for effective inhibition of phenoloxidase activity, a solution (or crystals) of phenylthiourea is/are used at the moment of hemolymph collection to prevent melanization.

Figures 3 and 6 – There are no error bars presented.

Line 303 – Please, adapt this citation to the journal rules.

Line 515 – Please, provide the authors names.

Author Response

Dear Sir/Madam,

Thank you for the thorough review of our paper and the opportunity to submit a revised version this second time. Again, we much appreciate the reviewer’s constructive comments on our manuscript.

Our responses are detailed below in red and yellow in the highlighted manuscript.

With the manuscript changes detailed below and our answers to the reviewer’s comments, we hope you will now find the revised version of our manuscript acceptable for publication.

Sincerely,

Dr. Eduard Torrents

Reviewer 2

Comments on the manuscript entitled: „A straightforward method for the isolation and cultivation of Galleria mellonella hemocytes” by Joana Admella and Eduard Torrents (Manuscript ID: ijms-1995811)

General comments

  1. The Authors cited only some selected papers published on in vitro insect hemocytes’ culture. And yet there are papers in which primary cultures of G. mellonella hemocytes have been successfully used and medium composition as well as culture conditions have been provided. Please, see e.g. Bogus et al. (Bull Entomol Res 2017, 107, 66-76), Kaczmarek et al. (Int. J. Mol. Sci. 2022, 23, 5204) and many papers published by this team. The Authors should at least compare their own findings with those published already on G. mellonella hemocytes’ primary cultures.

 We thank the reviewer for this observation. Please, see the added discussion on page 11, lines 336-34 taking into account these references and compared with our work.

  1. In vitro evaluation of the toxicity of various compounds and structures (e.g. nanoparticles) to eukaryotic cells can successfully be performed using mammalian cell lines. The use of in vitro cultured insect hemocytes for this purpose can present many problems (including interpretation of results) even when using primary hemocyte culture, mainly because they are very sensitive, labile cells that dye fast.

We appreciate the reviewer for this comment. We agree with the reviewer. The utility of our hemocytes cultures for this purpose also comes from the fact that after testing the compounds in vitro, we can go into the same in vivo animal model (Galleria mellonella) and perform more experiments in the same model. In contrast, we cannot do the same with mammals. Please, see the added sentence on page 2, line 79, discussing this point.

  1. There is no information about the performed statistical analysis of the results. 

We thank the reviewer for pointing this out. Concerning Table 1, the viability column summarizes 3-5 different experiments performed at the very beginning of our project. A deviation does not appear in the table, as Day 1 viability was calculated only as an estimation by observation. Please, see the added information on page 4, line 132.

About Figure 3, we apologize for missing the statistics before. These experiments were performed in triplicates. The graph and figure legend have been modified accordingly. Please, see on page 5 the new figure and figure legend 3.

In Figure 6, the experiment with the optimized protocol was performed twice. The statistics had not been added before due to the complexity of the plot being a double graph. A figure modification has been made, adding only the standard deviation bars in the line graph. The bar graph, which describes the number of cells, was quite similar, and we had to leave it like it was to make more accessible the visualization and understanding of the plot. Please, see on page 8 the new figure and figure legend 6.

In Figure 8, the values represented belong to one experiment performed in triplicates. The error bars indicate a positive standard deviation. Please, see the added information on figure legend 8, page 10, line 303.

All statistics analyses were performed with Graph GraphPad Prism 9.0 software. Please, see the information added as a sentence in section 4.3.2 (line 406) and section 4.5 (line 448).

Some other points

Line 19  - “… in an in vivo environment”? Is it correct?

We appreciate the reviewer for this observation. We have corrected this in the manuscript. Please, see the correction on line 22.

Line 88 and Line 338 - If the tails of the larvae were cut off for hemolymph collection, how was the contamination of the hemolymph with intestinal material avoided? The hindgut must have been broken when the tail was cut off. Could the presence of such contaminants, as foreign bodies for hemocytes, cause melanization and/or aggregation of hemocytes? Usually, the hemolymph is collected by incising (not cutting off) one of the last proleg.

We thank the reviewer for this interesting observation. Hemolymph can be collected, as the reviewer mentions, by incising. However, cutting the tails off is a common technique to extract hemolymph by letting the larva bleed. Cutting the larvae too much indeed makes it easier to disrupt the gut. However, cutting the larvae near the tail reduces the chance of disrupting the gut and contaminating the sample. In our group, we have wide experience extracting hemolymph with this method and never had any problems before. Our hemocytes cultures never got contaminated, but we cannot discard the possibility of some foreign bodies influencing the state of the cultures.

Please, see the addition of a new sentence explaining this point better. See on page 2, line 93.

Lines 148-149 – Why was L-cysteine added only during the suspension phase in the medium and not during the hemolymph collection? Usually, for effective inhibition of phenoloxidase activity, a solution (or crystals) of phenylthiourea is/are used at the moment of hemolymph collection to prevent melanization.

We thank the reviewer for pointing this out. Melanization mainly occurred once the cells were placed in the medium. Hemolymph was maintained on ice, specifically to prevent melanization during the collection and washing of hemocytes. For this, we decided to add the L-cysteine once the cells were in culture. This method completely avoided melanization, so we did not consider another option, which worked perfectly for us. Please, see the modifications explaining this point better on page 5, lines 153-156.

Figures 3 and 6 – There are no error bars presented.

We appreciate the reviewer for this observation. About Figure 3, we apologize for missing the statistics before. These experiments were performed in triplicates. The graph and figure legend have been modified accordingly. Please, see on page 5 the new figure and figure legend 3.

In Figure 6, the experiment with the optimized protocol was performed twice. The statistics had not been added before due to the complexity of the plot being a double graph. A figure modification has been made, adding only the standard deviation bars in the line graph. The bar graph, which describes the number of cells, was quite similar, and we had to leave it like it was to make more accessible the visualization and understanding of the plot. Please, see on page 8 the new figure and figure legend 6.

Line 303 – Please, adapt this citation to the journal rules.

We thank the reviewer for pointing this out. The citation has been adapted to the journal rules. Please, see the modification of reference 34 on line 315.

Line 515 – Please, provide the authors names.

We appreciate the reviewer for making this suggestion. The authors' names have now been provided. Please, see the modified reference (num 39) on line 545.
